# Interstitial Deletion of 3q21 in a Kuwaiti Child with Multiple Congenital Anomalies—Expanding the Phenotype

**DOI:** 10.3390/genes14061225

**Published:** 2023-06-05

**Authors:** Noor Almoosawy, Fawaz Albaghli, Haya H. Al-Balool, Hanan Fathi, Waleed A. Zakaria, Mariam Ayed, Hind Alsharhan

**Affiliations:** 1Department of Pediatrics, Faculty of Medicine, Kuwait University, P.O. Box 24923, Safat 13110, Kuwait; noor.almoosawy@hscm.ku.edu.kw; 2Department of Neonatology, Jaber Al-Ahmed Hospital, Ministry of Health, Hawalli 91712, Kuwait; 3Kuwait Medical Genetics Center, Ministry of Health, Ghanima Alghanim Building, Al-Sabah Medical Area, P.O. Box 5833, Hawalli 91712, Kuwait; 4Department of Pediatrics, Farwaniya Hospital, Ministry of Health, P.O. Box 13373, Farwaniya 81004, Kuwait; 5Radiology Department, Farwaniya Hospital, Ministry of Health, P.O. Box 13373, Farwaniya 81004, Kuwait; 6Department of Neonatology, Farwaniya Hospital, Ministry of Health, P.O. Box 13373, Farwaniya 81004, Kuwait; 7Department of Genetic Medicine, Johns Hopkins University School of Medicine, Baltimore, MD 21231, USA

**Keywords:** chromosome 3, interstitial deletion, microarray, congenital anomalies, dysmorphism

## Abstract

Interstitial deletions in the long arm of chromosome 3, although relatively rare, have previously been reported to be associated with several congenital anomalies and developmental delays. Around 11 individuals with interstitial deletion spanning the region 3q21 were reported to have overlapping phenotypes, including craniofacial dysmorphism, global developmental delay, skeletal manifestations, hypotonia, ophthalmological abnormalities, brain anomalies (mainly agenesis of corpus callosum), genitourinary tract anomalies, failure to thrive and microcephaly. We present a male individual from Kuwait with a 5.438 Mb interstitial deletion of the long arm of chromosome 3 (3q21.1q21.3) detected on the chromosomal microarray with previously unreported features, including feeding difficulties, gastroesophageal reflux, hypospadias, abdomino-scrotal hydrocele, chronic kidney disease, transaminitis, hypercalcemia, hypoglycemia, recurrent infections, inguinal hernia and cutis marmorata. Our report expands the phenotype associated with 3q21.1q21.3 while summarizing the cytogenetics and clinical data of the previously reported individuals with interstitial deletions involving 3q21, thus providing a comprehensive phenotypic summary.

## 1. Introduction

Few cases have previously been reported of deletions of the long arm of chromosome 3 involving 3q21 (Table 1) [1,2,3,4,5,6,7,8,9,10,11,12]. Recent advances in genetic analysis with high-resolution microarray techniques can play a crucial role in detecting chromosomal abnormalities not previously detected in conventional cytogenetic studies, thus leading to revolutionized genetic diagnosis. Chromosomal microarray analysis (CMA) is considered first-tier testing in evaluating disorders related to intellectual disabilities, autism, and/or multiple congenital anomalies [13]. Here, we report a male individual with an interstitial deletion of the long arm of chromosome 3 (3q21.1q21.3) with multiple congenital anomalies and previously unreported clinical findings that further expand the clinical spectrum of 3q21 deletions. The deleted region contains about 104 genes, 38 of which are OMIM genes. While the functions of most genes are yet to be elucidated, around six genes are known to be associated with autosomal dominant (AD) diseases, which include *MCM2*, *ZNF148*, *SEC61A1*, *RAB7A*, *MYLK*, and *GATA-2* (Appendix A). The early molecular diagnosis of our case not only enabled proper management but also led to screening and, thus, prevention of other potential complications associated with 3q21 deletion. Our report summarizes the molecular and clinical data of the 13 previously reported individuals with 3q21 deletions (Table 1). We believe that the following are the key genes responsible for the overlapping phenotype: *ZNF148*, *SEC61A1*, and *GATA-2.*

**Table 1 genes-14-01225-t001:** Overview of individuals with chromosome 3q interstitial deletions, including regions q13 to q23. ACC: agenesis of the corpus callosum; B/L: bilateral; CC: corpus callosum; F: female; FT: full term; GERD: gastroesophageal reflux disease; HC: head circumference; Hr: hour; Ht: height; ID: intellectual disability; IDA: iron-deficiency anemia; M: male; mo; month; NA: not applicable; ND: not determined; PDA: patent ductus arteriosus; PFO: patent foramen ovale; UPJ: ureteropelvic junction; SD: standard deviation; VSD: ventricular septal defect; VUR: vesicoureteral reflux; Wk: week; Wt: weight; Yr: year; %: percentile.

	Our Case	Arai et al., 1982 [10]	Jenkins et al., 1985 [1]	McMorrow et al., 1986 [2]	Okada et al., 1987 [7]	Fujita et al., 1992 [6]	Genuardi et al., 1994 [11]	Wolstenholme et al., 1994 [3]	Ogilivie et al., 1998 [9]		Callier et al., 2009 [8]	Molin et al., 2012 [4]	Materna-Kiryluk et al., 2014 [5]	Vinh et al., 2018 [12]
									**Case 1**	**Case 2**				
**Chromosomal deletion**	3q21.1q21.3	3q13q21	3q11q21	3q12q21	3q12q21	3q12q23	3q13.12q21.3	3q21q23	3q12q21	3q12q21	3q21.1q21.3	3q13.32q21.2	3q13.31q22.1	3q21.3
**Sex**	M	M	F	M	F	M	M	M	F	M	F	M	F	M
**Age at report**	2-yr-11-mo	Neonate (58 hr)	7-yr	Infant (age ND)	8-yr	6-yr	5-mo	Fetus	4-yr	17-mo	14-yr	8-yr 10-mo	14-yr	17-yr
**Gestational age**	31-wk	41-wk	FT	ND	40-wk	40-wk	ND	25-wk	FT	37-wk	FT	FT	35-wk	28-wk
**Failure to thrive (Wt<2nd%;-2SD)**	+	+	-	ND	+ (Wt & Ht <1st%)	+ (Wt -2SD)	-	-	-	-	+ (Ht—4SD) +(wt—10th centile)	-	+ (<2nd %)	-
**Developmental delay/ID**	+	+	+	ND	+	+	ND	NA	+	+	+	+ (speech)	+	-
**Hypotonia**	+	ND	+	ND	+	+	+	NA	+	+	-	-	+	-
**Microcephaly (HC** **≤** **-2SD)**	+ (-3SD)	+	-	ND	+ at birth (-3SD)	+ (HC -2SD)	-	-	-	-	-	-	-	-
**Seizures**	+	ND	ND	ND	ND	ND	ND	ND	ND	ND	ND	ND	+	ND
**Brain anomalies**	+ (Focal calcified focus in right parietal white matter, periventricular leukomalacia, dysplastic CC)	+ (alobar holoprosencephaly)	-	+ (dilated ventricle, ACC)	ND	-	+ (ACC)	-	-	+ (ACC)	+ (ACC)	ND	+ (ACC), cortical atrophy	-
**Craniofacial dysmorphism**	Wide anterior fontanelle, wide-spaced sutures, broad forehead, hypertelorism, epicanthal folds, broad and flattened nasal bridge, anteverted nares, pointed chin, small mouth, down-turned mouth angles, high-arched palate	Mid-facial dysplasia, midline cleft lip and palate	Brachycephaly, large head, epicanthal folds, antimongoloid slant	Dysmorphic facies	Plagiocephaly, hypertelorism, epicanthal folds, broad/flat nasal root, long pointed chin, high arched palate, mid-facial dysplasia, asymmetrical face	Flat occiput, low nasal bridge, long flat face, large odd-shaped ears, thin lips, epicanthal folds, narrow and short palpebral fissures, fixed facial appearance	Dolicocephaly, large head, bitemporal narrowing, mild cranial asymmetry, hypertelorism, epicanthal folds, broad nasal root, anteverted nares, low-set ears, posteriorly rotated, long philtrum, small mouth	Shortened palpebral fissures, no upper eyelids folds, beaked nose, telecanthus	Broad face, high arched palate, pointed chin, large head	Plagiocephaly, broad forehead, mid-face hypoplasia, prominent supra-orbital ridges, upturned nose, prominent nostrils, antimongoloid slant to the eyes, hypertelorism, over-folded ears, pointed chin, high arched palate	Right coronal craniosynostosis, asymmetrical face, low-set ears, hypoplasia of right ear, left pre-auricular tags, Down slanting palpebebral fissures	Epicanthal folds, hypertelorism, antimongoloid slant	Macrostomia, asymmetrical narrow palpebral fissures, right pre-auricular tags	-
**Ophthalmological anomalies**	-	ND	+ (Ptosis, pterygium)	ND	ND	+ (Blepharophimosis, ptosis, epicanthus inversus, nuclear cataract, exotropia, nystagmus)	+ (Nystagmus)	+ (Blepharophimosis, ptosis, epicanthus inversus)	+ (Myopia)	ND	Strabismus	+ (Ptosis, strabismus)	+ (Blepharophimosis, myopia)	-
**Skeletal/limbs findings**	-	ND	+ (Moderate thoracic kyphosis, excessive lumbar lordosis, broad chest)	+ (Small thoracic cage, talipes equinovarus, joint contractures)	+ (Scoliosis, multiple joint contractures, talipes equinovarus)	+ (Scoliosis, joint contractures, pes valgus, talipes equinovarus, short sternum, funnel chest, long tapered fingers, bilateral simian crease, whorls on all fingers) Ataxic gait at age of 6	+ (Joint contracture, talipes equinovarus, tapered fingers, adducted thumbs, deviating 2nd toe bilaterally)	-	Unusual palmar crease	Broad hands and feet, scoliosis, lordosis, kyphosis, hypoplastic pelvis	+ (Pes planus, genu valgum, duplication of right thumb, short stature)	+ (Small hands with bilateral single palmar crease)	Phalangeal hypoplasia, hypoplastic vestigial left thumbs, absent radial bones, left clubfoot varus, left foot syndactyly (2nd & 3rd toes), scoliosis, joint contractures, short stature	-
**Skin manifestations**	+ (Cutis marmorata telangiectasia)	ND	+ (Reddish skin of hands and feet)	ND	+ (Freckle-like pigmentations on face and forearms)	ND	Puffy feet (edema)	ND	ND	ND	ND	ND	ND	Waxing and waning lower limb nonpitting edema, lower extremity warts
**Genitourinary malformations**	+ congenital bilateral hydronephrosis, grade IV VUR, hypospadias (megameatus intact prepuce), cryptorchidism of left undescended testis, corrected, right abdomino-scrotal hydrocele, right pyeloplasty with DJ stent, Right UPJ obstruction, chronic kidney disease stage IV), corrected left inguinal hernia (24/6/2020). DMSA scan showed renal scarring and impaired renal function	+ (Hypoplastic penis, cryptorchidism)	-	B/L cryptorchidism, unilateral hydronephrosis	+ (Right incomplete duplicated collecting system)	+ (Cryptorchidism)	+ (Hypoplastic penis, cryptorchidism, ureteral enlargement, urethral valve stenosis)	-	-	-	-	-	+ (Right renal agenesis, grade IV left VUR, left-sided hydronephrosis, prominent clitoris)	Bilateral vesico-ureteral reflux with left hydronephrosis requiring bilateral ureteral reimplantation at 6 years
**Hematological abnormalities**	Recurrent neutropenia, anemia	ND	ND	ND	IDA, excess IgG	ND	ND	-	ND	ND	Pancytopenia, myelodysplasia	ND	ND	Pancytopenia, hypocellular marrow
**Cardiac malformations**	+ (PFO, trivial tricuspid regurgitation)	PDA	Heart murmurs (+PFO, no significant heart disease)	ND	-	-	ND	-	ND	ND	PDA	ND	PDA, PFO, VSD, aortic coarctation, bicuspid aortic valve	-
**Other**	Feeding difficulties, GERD, G-tube depended, hypercalcemia, mild transaminitis, hypoglycemia, apneic episodes, left inguinal hernia, recurrent infections	Hirsutism	Puffy hands and feet lymphedema, mildly webbed neck, wide-spaced nipples, hiatal hernia, recurrent aspiration pneumonia	Fibroblasts showed same deletion. Parents had normal karyotype.	Type II fiber atrophy on muscle biopsy	Severe constipation	Wide-spaced nipples	Absent left hemidiaphragm	Poor feeder, broad based gait	Apneic episodes	Unilateral deafness due to ear canal stenosis, chronic constipation, hyperactivity, self-aggression, precocious puberty		Left thyroid lobe hypoplasia, hypothyroidism	

## 2. Case Report

We report a boy aged 2-years-and-11-months-old who was born to consanguineous parents at 31-week gestation through a lower segment caesarian section due to maternal pre-eclampsia. The mother had two first-trimester miscarriages after this pregnancy, neither of which were investigated. The birth weight of the boy was at the 14th percentile. Congenital renal anomalies were diagnosed pre-natally and confirmed post-natally to include congenital bilateral hydronephrosis (grade II in the left kidney and grade IV in the right kidney), bilateral pelvi-ureteric junction (PUJ) obstruction, bilateral vesicoureteral reflux (VUR) (grade IV), hypospadias (mega meatus intact prepuce), left inguinal hernia, right abdomino-scrotal hydrocele, cryptorchidism, and patent foramen ovale (PFO) with a left-to-right shunt (Figure 1C,D). His post-natal period was complicated by a prolonged neonatal intensive care unit (NICU) stay that lasted for a couple of months due to his prematurity, congenital renal anomalies, and respiratory distress that required mechanical ventilation for 1 week. He underwent multiple surgical procedures between the ages of 2 and 15 months. The procedures included left inguinal herniotomy, left orchidopexy, and repair of hypospadias via the dorsal slit procedure. In addition, a right nephrostomy with pyloroplasty was repeated multiple times due to the worsening of hydronephrosis. A retrocaval ureter was also found at the time of surgery that was trans-positioned with a pyloroplasty procedure. He suffered repeated episodes of pyelonephritis that led to renal scarring and subsequent chronic kidney disease (CKD) and hypertension. Apneic episodes were also frequent in the first year of life, requiring multiple hospital admissions. The boy was noted to have microcephaly, failure to thrive (FTT) (weight, height and head circumference were < 1st percentile at 12-month-old), severe gastroesophageal reflux disease (GERD), and feeding difficulties requiring tube feeding (gastrostomy tube) and anti-reflux medications. Furthermore, he is also noted to have cutis marmorata, hypotonia, and dysmorphic features, which include a broad forehead, hypertelorism, epicanthal folds, depressed and wide nasal bridge, short nose, anteverted nares, downward mouth angles, and small chin (Figure 1A,B). He has global developmental delay; aged 12 months, he was able to fix, track, and turn to sounds and startle to loud noises. However, he did not pronounce any words and showed delayed development of head control with severe head lag and generalized hypotonia. Routine laboratory studies revealed progressive CKD stage IV with consistently elevated creatinine and urea; aged 2-years-and-8-months, creatinine was elevated at 240 (reference range (RR): 35–62 μmol/L), and urea was elevated at 17.4 (2.8; RR: 1.8–6.4 μmol/L). He has electrolyte abnormalities, mainly hypercalcemia (2.62; RR 2.17–2.44 μmol/L), elevated magnesium at 1.04 (RR: 0.62–0.95 mmol/L), and elevated phosphorus at 2.24 (RR: 1–1.95 μmol/L), as well as high levels of the parathyroid hormone at 13.6 (RR: 1.3–9.3 μmol/L) and normal vitamin D level. Urinary studies included normal urine pH (5.5); however, urine electrolytes were not obtained. He had hyperkalemia and persistent non-anion gap metabolic acidosis, which was suggestive of distal renal tubular acidosis type IV, which was managed with oral sodium citrate supplementation. He also has frequent infections involving the respiratory and urinary tracts, which might be explained by his recurrent mild neutropenia ranging from 0.7–1.4 × 10^9^ (RR: 1.5–8 × 10^9^). Immunological workup revealed normal immunoglobulin levels (IgM, IgG) and mildly elevated IgA at 1.15 (RR: 0.2–1 g/L). He has normocytic normochromic anemia; hemoglobin is also reduced at 10.4 g/dl (RR: 11–14). He was also found to have mild transaminitis, which was characterized by ALT 67 (RR:10–60 IU/L) and AST 136 (RR: 10–42 IU/L), as well as high serum cholesterol (5.53 mmol/L; RR: 3.1–5.2 mmol/L) and triglycerides (2.26; RR: 0–1.7 mmol/L). His thyroid function was within normal limits. He had recurrent transient episodes of hypoglycemia aged 11 months old, which occurred during one of his hospitalizations. This condition was attributed to inadequate feeding as he had unremarkable routine metabolic testing, including plasma amino acids, acylcarnitine, ammonia, lactate, insulin, growth hormone, and urine organic acids, in addition to his normal newborn screening. Brain magnetic resonance imaging (MRI) performed at 4 months of age showed an altered signal in the periventricular white matter and dysplastic corpus callosum that was thinning out posteriorly (Figure 1E). Brain computed tomography, which was obtained at 11 months of age, showed a three-mm calcified focus within the right parietal deep white matter, with no mass effect. A repeat brain MRI at the age of 15-month revealed bilateral, slightly asymmetrical periventricular and deep white matter gliotic scarring, which was consistent with periventricular leukomalacia. Abdominal ultrasound at 14 months of age showed bilaterally enlarged kidneys with marked hydronephrotic changes, as well as increased renal parenchymal echogenicity with right-sided DJ catheter in place with unremarkable liver and spleen. He was diagnosed with epilepsy with generalized seizures at 12 months of age, for which he is well controlled on a single antiepileptic drug (Keppra). He had normal audiology more recently. He continues to express signs of developmental delay; however, he has acquired new developmental milestones with improvement in muscle tone, and, due to ongoing physical therapy and adequate nutrition, he was able to sit unsupported at 18 months. Nonetheless, he continues to be non-verbal and unable to stand or cruise at 2-years-and-11-months old. He has not had a formal neuropsychology evaluation yet due to his frequent hospitalizations. Since the treatment of his condition is largely supportive, none of the symptoms could be attributed as iatrogenic.

## 3. Chromosome Microarray Analysis

Genomic DNA was isolated from peripheral blood using QIAsymphony fully automated DNA extraction instrument (Qiagen, Hilden, Germany) according to the manufacturer’s protocol. DNA concentration and purity was checked using NanoDrop 2000/2000c spectrophotometers (ThermoFisher Scientific, Waltham, MA, USA) according to the manufacturer’s protocol. Chromosomal microarray was carried out using the Affymetrix Cytoscan HD Microarray assay, which containd approximately 1.9 million copy number oligonucleotide probes and approximately 750,000 SNP probes (Affymetrix, Santa Clara, CA, USA). The procedure was followed as described in [13]. Using the Chromosomal Analysis Suite (ChAS) software (ThermoFisher Scientific), the sample results were analyzed at a resolution of 50 kb for copy number losses and 200 kb for copy number gains across the genome. The results revealed arr[hg19]3q21.1q21.3(123413267_128851482) × 1 male. The array showed a copy number loss involving the long arm of chromosome 3, which was located from band 3q21.1 to q21.3 with a size of 5.438 megabase (Mb). The deleted region contained about 104 genes, 38 of which were OMIM genes where 6 genes are associated with AD phenotypes (Figure 1F; Appendix A). Following the ACMG guidelines for copy number variant interpretation and classification [13], this aberration was classified as likely pathogenic in nature. Parental genetic testing could not be obtained.

## 4. Discussion

Although most of the cases with interstitial deletion involving 3q21 previously reported in the literature (Table 1) had an overlapping phenotype, a clear genotype–phenotype relationship was not determined, given the limited number of affected individuals with the exact chromosomal interstitial deletion [1,2,3,4,5,6,7,8,9,10,11,12]. There are about 13 cases reported with interstitial deletions involving region 3q21 (Table 1). The CMA of our case revealed a 5.438Mb interstitial deletion of (3q21.1q21.3), affecting about 104 genes, 38 of which were OMIM genes, while only 6 genes were known to be associated with AD-known disorders (Appendix A). The 5.4Mb deletion within 3q21.1q21.3 found in this study is submicroscopic interstitial deletion that was detected using array comparative genomic hybridization. Such deletion can occur due to different mechanisms, including DNA recombination, replication or repair associated processes. For example, non-allelic homologous recombination between low-copy repeats can create deletion, as it was reported in 22q11.2 region that cause DiGeorge syndrome [14]. Two possible scenarios for this deletion, inter-chromosomal recombination takes place between two chromosome regions with >99% identity. The second scenario is intra-chromosomal recombination crossing over within the same chromosome [14]. Both of which will result in the loss of the corresponding region. Examining the breakpoint of such a deletion could help in further understanding the mechanism of this deletion. This can be carried out in the future by sequencing the breakpoint in the proband using a series of polymerase chain reactions, or it can also be achieved by whole genome sequencing.

In addition to the reported congenital anomalies of the kidney and urinary tract (CAKUT) in individuals with 3q21 interstitial deletions, such as hypoplastic penis, prominent clitoris, cryptorchidism, urethral valve stenosis, vesicoureteral reflux, hydroureter, duplicated collecting system, renal agenesis, and hydronephrosis, our case has additional unreported CAKUT, which include abdomino-scrotal hydrocele; hypospadias (mega meatus intact prepuce); ureteropelvic junction obstruction, resulting in severe progressive hydronephrosis; and CKD stage IV. As a result of the obstructive uropathy and subsequent renal insufficiency, he developed secondary hyperparathyroidism, as is evident in his biochemical abnormalities: elevated PTH levels, moderate hypercalcemia, hypermagnesemia, and elevated serum phosphorus. Calcification of the parietal deep white matter observed in our case might be a consequence of his hypercalcemia. He has further previously unreported features, such as cutis marmorata, transaminitis, recurrent infections with neutropenia, inguinal hernia, feeding difficulty, GERD, and FTT requiring gastrostomy. Moreover, he had mild transaminitis, hypercholesterolemia, hypertriglyceridemia, and recurrent but transient episodes of hypoglycemia. There were multiple previously reported Decipher cases with 3q deletions (Appendix A); however, there are two main cases that overlap most clearly with our subject in the deleted 3q21.1q21.3 coordinates: 123413267–128851482 (Appendix A). Case 252244 involved a 6-year-old female with a pathogenic de novo 4.87 Mb deletion in chromosome 3 (coordinates:123633665–128506246), who was reported to have microcephaly, short stature, and intellectual disability [15]. The other case, i.e., 274735, involved a 7-year-old male with a de novo 4.12 Mb deletion in chromosome 3 (coordinates 123522445–127638835), who was reported to have an intellectual disability [15].

The deleted region in our subject harbors one of the Krüppel-type zinc finger genes (*ZNF148*), from which heterozygous loss of function variants are linked to intellectual disability syndrome disorders associated with autism spectrum disorder (ASD), attention deficit hyperactivity disorder (ADHD), developmental delay, underdevelopment of corpus callosum (CC), short stature, feeding difficulty, variable microcephaly or mild macrocephaly, facial dysmorphism (hypertelorism, a low columella and pointed chin), and cardiac and renal manifestations (hydronephrosis, multicystic dysplastic kidneys). Different de novo heterozygous *ZNF148* truncating (nonsense or frameshift) variants were reported in five unrelated individuals all located in the last exon of *ZNF148*, resulting in a premature termination codon and, presumably, truncated proteins [16,17]. Despite the genotypic similarities among these five cases, they presented significant phenotypic variability. To our knowledge, this study presents the first case with deletion in *ZNF148*. The reported phenotype of *ZNF148*-related syndrome is consistent with most of the clinical findings observed in our case, except for the ASD and ADHD, which are premature to diagnose in our subject. *ZNF148* is involved in transcriptional regulation, with an unclear role in the neurological development. However, it is thought to be a key regulator of *PKD1* and *PKD2* gene promotors, which could explain the renal anomalies observed in 2/5 reported individuals in whom its function is lost [16,18]. In addition to *ZNF148* deletion, CKD in our subject could also be explained based on the deletion of *SEC61A1*, which is associated with AD progressive tubulointerstitial kidney disease (ADTKD) characterized by tubular damage and interstitial fibrosis without glomerular involvement, resulting in CDK and end-stage renal disease in adulthood [19,20]. Heterozygous missense variants in *SEC61A1* were reported in two families with syndromic progressive CKD with small, dysplastic kidneys associated with cysts, congenital anemia, intrauterine, or post-natal stunted growth and neutropenia [20,21]. Two additional affected yet unrelated families with heterozygous missense and nonsense variants in *SEC61A1* were described as having defects in plasma cell development, hypogammaglobulinemia, and early-onset recurrent bacterial infections that responded well to immunoglobulin replacement therapy [21], unlike our case, in which the patient had normal immunoglobulin levels but recurrent early-onset bacterial and viral infections. Another important gene located in the deleted region is *GATA2*, which regulates the function of hematopoietic stem cells and is linked to Emberger syndrome (lymphedema) (MIM#614038), immunodeficiency 21 (MIM#614172), and susceptibility to acute myeloid leukemia (MIM#601626) and myelodysplastic syndrome (MDS) (#MIM614286), which recently became referred to as *GATA2* deficiency syndrome [8,12,22,23,24]. It is associated with AD predisposition to hematological and non-hematological disorders, namely myeloid malignancy (in 75% of *GATA2* carriers), and immunodeficiency with recurrent infections (including viral, fungal, and bacterial) (in 62% of *GATA2* carriers) attributed to various cytopenias, respectively [24]. A majority of identified germline *GATA2* variants were loss-of-function-related, with gene deletions reported in 15.6% of cases [24]. The age of onset of myeloid malignancy ranges from 0–78-years-old, with a median of 17-year-old [24]. Furthermore, in a cohort of children and adolescents with MDS, germline *GATA2* variants were found to account for 15% of advanced and 7% of all primary pediatric MDS cases [23]. In addition to hematologic phenotype, *GATA2* haploinsufficiency is linked to deafness, as well as urogenital tract anomalies, autism, and aggressive behavior [23]. Our case continued to have normal complete blood counts (CBC) with differentials at the age of 27 months and showed no signs of lymphedema. However, he suffers recurrent infections that could be attributed to his neutropenia. Given the impact of germline *GATA2* deletion, surveillance for myeloid malignancy was initiated in our subject to ensure timely and adequate intervention. We recommend obtaining CBC with differentials every 6 months. Heterozygous pathogenic missense variants in *MCM2* were linked to AD non-syndromic, progressive sensorineural hearing loss (SNHL) in two unrelated families, with the variable age of onset ranging from 9 to 88 years old; the recorded patients also presented different levels of severity [25,26]. This knowledge led us to carry out regular hearing screening tests in our subject, which became normal at the age of 24 months. Two more genes deleted in our subject, which are associated with late-onset AD inheritance, are *RAB7* and *MYLK* genes. *RAB7* is associated with peripheral neuropathy, i.e., AD Charcot–Marie–Tooth type 2B (CMT2B), and a variable age of onset, which is typically early adulthood [27,28,29]; variation in the age of onset is another potential clinical finding that needs to be monitored and screened for in our index. *MYLK* gene is associated with thoracic aortic aneurysms and aortic dissection (TAAD) with incomplete penetrance and later onset in heterozygous individuals [30]. Echocardiogram showed normal aorta in our case; however, routine surveillance with time is warranted.

Our subject has additional unreported biochemical features, namely transient hypoglycemia, transaminitis, hypertriglyceridemia, and hypercholesterolemia, which could not be fully explained based on either his clinical history or the detected chromosomal abnormality. While most of the literature reported de novo microdeletions of 3q21, parental testing, in our case, could not be obtained to determine its inheritance. Although the information in the literature may propose that the deletion is possibly pathogenic, the cumulative effect of all genes in the region, not individual genes, is expected in such chromosomal large deletions. 

## 5. Conclusions

To conclude, our case further expands the phenotype associated with distal chromosome 3 microdeletion and emphasizes the role of chromosomal microarray analysis in explaining and delineating multiple associated congenital anomalies. We have provided an extensive summary of previously reported cases. Early molecular diagnosis in this individual positively impacted the clinical care and allowed surveillance for several potential disorders, thus preventing future disease risk associated with 3q21 interstitial deletion, highlighting the significance of timely genetic testing. 

## Figures and Tables

**Figure 1 genes-14-01225-f001:**
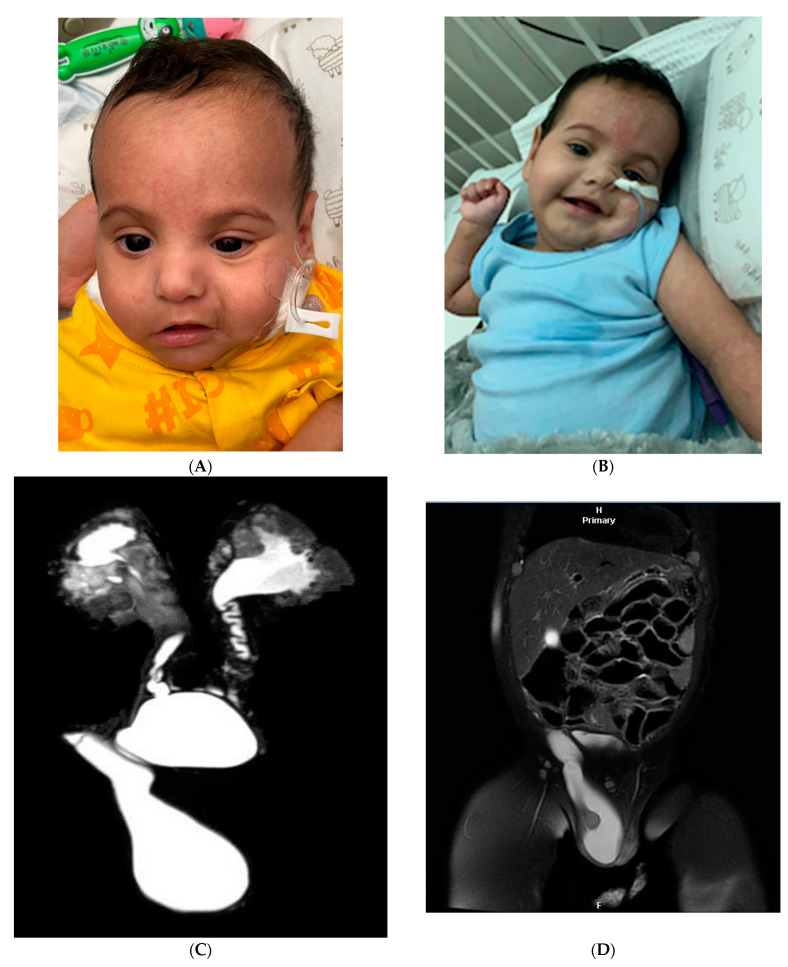
(**A**,**B**) Photography of our subject showing dysmorphic features. Note dysmorphic features: broad forehead, hypertelorism, epicanthal folds, depressed and wide nasal bridge, short nose, anteverted nares, small chin, and downward mouth angles. Note cutis marmorata: (**C**,**D**) MRI urography demonstrating right inguinoscrotal hernia and bilateral vesicoureteral reflux; (**E**) brain magnetic resonance imaging (MRI) of subject aged 4 months old demonstrating dysplastic corpus callosum with thinning out posteriorly (arrow); (**F**) upper section shows deleted region (3q21.1q21.3)x1 using ChAS software (ThermoFisher Scientific, Waltham, MA, USA), which is located in long arm of chromosome 3 (chr3:12,3413,267–128,851,482 GRCh37/hg19). Bottom section shows 68 genes in this region (obtained from UCSC database).

## Data Availability

Not applicable.

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
