# Peer review of "Interstitial Deletion of 3q21 in a Kuwaiti Child with Multiple Congenital Anomalies—Expanding the Phenotype"

_genes, 2023, doi:10.3390/genes14061225_

Round 1

Reviewer 1 Report

1- Line 29, The sentence can be written as “Few cases have been reported with deletions of the long arm of chromosome 3 involving the 3q21 band”.

2- Gene abbreviations should be given in italic form (MCM2).

3- Karyotype nomenclatures should be written according to ISCN 2020.

4- As in this case, although databases report it as a possible pathogenic, the cumulative effect of all genes in the region, not individual genes, is expected in chromosomal large deletions. Therefore, this effect should be included in the discussion.

5- Was chromosome analysis performed after the microarray study, which is visible when the deletion is targeted? In addition, it needs to be examined for possible insertional translocations in the mother and father. Because there are pregnancy losses for which the cause has not been explained.

Author Response

Chromosome 3q21 Review Comments

RESPONSE TO THE COMMENTS OF THE REVIEWERS:

Reviewer's Comments:
Reviewer: 2

Thank you for valuable feedback and for considering our manuscript for publication at Genes.

Line 29, The sentence can be written as “Few cases have been reported with deletions of the long arm of chromosome 3 involving the 3q21 band”.

This was corrected.

Gene abbreviations should be given in italic form (MCM2).

This was corrected.

Karyotype nomenclatures should be written according to ISCN 2020.

WE have edited that. Here is what we corrected under “Chromosomal microarray analysis”:

arr[hg19]3q21.1q21.3(123413267_128851482)x1 male

As in this case, although databases report it as a possible pathogenic, the cumulative effect of all genes in the region, not individual genes, is expected in chromosomal large deletions. Therefore, this effect should be included in the discussion.

This has been added as recommended.

 Was chromosome analysis performed after the microarray study, which is visible when the deletion is targeted? In addition, it needs to be examined for possible insertional translocations in the mother and father. Because there are pregnancy losses for which the cause has not been explained.

For our patient, we did not perform any further chromosomal analysis apart from the chromosomal microarray. We have tried to obtain the parental testing, however the probes for the region3q21.1q21.3 is unavailable at our local lab. We plan to obtain parental subtelomere probes as well as SNP array. However, parents are not willing to do testing.

Reviewer 2 Report

Reports of patients with contiguous gene deletion are very important because can aid to explain the functions of some genes that are not so clear in literature and give some important information about follow-up. 

In this report, the role of single genes in the phenotype could be improved because is not well described and it is proposed an expansion of the phenotype associated with distal chromosome 3 microdeletion without have excluded other clinical differential diagnoses with other diagnostic exams of their patient.  A description of the other OMIM genes and their likely association with the patient's clinical picture is missing. It's important to control the deleted region and the OMIM genes that are included. 

lines 24-25-26: delete

lines 41-42:  the list of OMIM genes that you mention is not correct. ADCY5 AND CASR are not included in your deletion (123413267_128851482). 

ADCY5 is located in 123,001,143-123,167,392

CASR is located in 121,903,181-122,005,344

there are also other OMIM genes that you didn't mention (see https://genome.ucsc.edu/cgi-bin/hgTracks?db=hg19&lastVirtModeType=default&lastVirtModeExtraState=&virtModeType=default&virtMode=0&nonVirtPosition=&position=chr3%3A123413267%2D128851482&hgsid=1607743653_kVluf9kzD8UiOsP1x2uJBVLx7AH2). 

 lines 64-65: you could explain better why the baby went to NICU...respiratory distress? prematurity?

lines 79-80: he didn't pronounce any words and showed delayed development of Head Control

lines 82-83: better explain kidney function. You didn't mention other urinary electrolytes. What is the clinical renal condition that was suspected? 

line 88: explain if neutropenia could justify the recurrent infections.

line 91: what is abdomen echography? it was normal? That was liver hypertrophy? there was a neuromuscular disease associated? how is CPK?

line 94: when? neonatal period or after? how is the insulin level?

line 94-96: change resolved with investigated with newborn screening and routine metabolic testing, including plasma amino acids, acyl-carnitine, urine organic acids, ammonia, and lactate that were unremarkable. 

line 104: focal? generalized?

line 105: data from the last follow-up examination, age, auxological data and clinical situation, psychomotor developmental milestones achieved are missing

line 112: from the photographs seem cutis marmorata without teleangiectasia and in the text you mentioned only cutis mamorata

lines141-142: review the number of UCSC genes and OMIM genes. 

line 142: add DECIPHER case and his phenotype. 

line 144: actually other Cakut alterations have already been described in the other patients, and these types of issues are heterogeneous even in the same family. 

line148: define FTT

lines 148-149: Additionally, he showed some unique metabolic abnormalities. Define and better explain. 

lines 148-158: CASR is not included in your deletion! 

chr3:121,903,181-122,005,344 

hg 19/GRCh37

this is region of casr and it's outside of your deleted region indicated in the text (123413267_128851482, hg19)

lines 159 and 167 : ZNF187, change in ZNF148

line163: it's necessary to explain the different types of variants described in Stevens et al. There are also few patients described and so there are limited information. 

line167: explain why znf148 deletion can explain kidney disease. What are the kidney problems described in the patients of stevens et al? how many patients?

line 176: there are more recent reports like this Homan CC, Venugopal P, Arts P, Shahrin NH, Feurstein S, Rawlings L, Lawrence DM, Andrews J, King-Smith SL, Harvey NL, Brown AL, Scott HS, Hahn CN. GATA2 deficiency syndrome: A decade of discovery. Hum Mutat. 2021 Nov;42(11):1399-1421. doi: 10.1002/humu.24271. Epub 2021 Aug 31. PMID: 34387894; PMCID: PMC9291163. Compare and comments the other gata2 patients in the literature and  and also comment the cancer risk. have you proposed to your patient a cancer surveillance? 

lines 179-181: to define a unique features never described before and to hypothesize that they are caused by the deletion, it is necessary to analyze all the clinical differential diagnoses. have you tested Insulinemia? what hepatic echography was? was there familiar hypercholesterolemia? it was caused by food?

line181: add the other OMIM genes in the deleted region and add comments.

lines 181: suggest surveillance and exams for future risk. You have not explained what are the clinical problems described in the other patients in the literature and not your patients. 

line 250: Stevens SJ and not S.SJ. 

let's control the bibliography. 

Table1:

-you can use abbreviations of years, months and days.

- It's not clear who is the last column (patient 2 of Vinh et al)?

-modify the "failure to thrive" part. You have to use or centile or %

-microcephaly: the patient of Okada doesn't have microcephaly. He presents HC at -0.05 SD. Microcephaly is defined as head circumference ≤-2SD 

Supplementary file: include only the right OMIM genes

Author Response

Reviewer's Comments:
Reviewer: 1

Reports of patients with contiguous gene deletion are very important because can aid to explain the functions of some genes that are not so clear in literature and give some important information about follow-up. 

In this report, the role of single genes in the phenotype could be improved because is not well described and it is proposed an expansion of the phenotype associated with distal chromosome 3 microdeletion without have excluded other clinical differential diagnoses with other diagnostic exams of their patient.  A description of the other OMIM genes and their likely association with the patient's clinical picture is missing. It's important to control the deleted region and the OMIM genes that are included. 

Thank you for valuable feedback and for considering our manuscript for publication at Genes. We have explained all the reported autosomal dominant phenotype of the genes deleted and we have related them to our case.

lines 24-25-26: delete

We have deleted these lines.

lines 41-42:  the list of OMIM genes that you mention is not correct. ADCY5 AND CASR are not included in your deletion (123413267_128851482). 

ADCY5 is located in 123,001,143-123,167,392

CASR is located in 121,903,181-122,005,344

Thanks for this correction. We have deleted these two genes in the manuscript and in the supplementary table.

there are also other OMIM genes that you didn't mention (see https://genome.ucsc.edu/cgi-bin/hgTracks?db=hg19&lastVirtModeType=default&lastVirtModeExtraState=&virtModeType=default&virtMode=0&nonVirtPosition=&position=chr3%3A123413267%2D128851482&hgsid=1607743653_kVluf9kzD8UiOsP1x2uJBVLx7AH2). 

We have added and correlated to our case accordingly. We have only included the six genes with autosomal dominant inheritance.

 lines 64-65: you could explain better why the baby went to NICU...respiratory distress? prematurity?

He was admitted to NICU for his prematurity, respiratory distress and congenital renal anomalies. This has been added to the manuscript.

lines 79-80: he didn't pronounce any words and showed delayed development of Head Control

This was corrected and edited.

lines 82-83: better explain kidney function. You didn't mention other urinary electrolytes. What is the clinical renal condition that was suspected? 

He was born with congenital renal urinary tract anomalies in the form of bilateral pelvi-ureteric junction obstruction and vesicoureteric reflux, he suffered repeated attacks of pyelonephritis that led to renal scarring and subsequent CKD and hypertension.

As for urinary studies, urine PH was 5.5 (he can acidify urine), urine electrolytes were not requested, he had persisted metabolic acidosis and hyperkalemia, with normal anion gap suggesting type IV distal RTA. This has been added to the manuscript.

line 88: explain if neutropenia could justify the recurrent infections.

Yes, we have added this sentence in the manuscript: “He has recurrent mild neutropenia ranging from 0.7-1.4 x109 (RR: 1.5-8 X109) that could explain the recurrent infections”.

line 91: what is abdomen echography? it was normal? That was liver hypertrophy? there was a neuromuscular disease associated? how is CPK?

Ultrasound abdomen was performed as mentioned in line 102 and was remarkable for renal anomalies as mentioned with normal liver and spleen (this has been added to the manuscript). There is no neuromuscular disease suspected. The hypotonia has improved with better nutrition and physical therapy. This information has been added. CPK has not been measured.

line 94: when? neonatal period or after? how is the insulin level?

He had hypoglycemia episodes during his hospitalization at age 11-month that was resolved. His metabolic work up was unremarkable including insulin. We have added this information.

line 94-96: change resolved with investigated with newborn screening and routine metabolic testing, including plasma amino acids, acyl-carnitine, urine organic acids, ammonia, and lactate that were unremarkable. 

The sentence has been edited.

line 104: focal? generalized?

Generalized. This has been added.

line 105: data from the last follow-up examination, age, auxological data and clinical situation, psychomotor developmental milestones achieved are missing

We have added more information about his development at the end of the paragraph, as follow: “He continued with developmental delay however, he has acquired new developmental milestones with ongoing physical therapy; he was able to sit unsupported at 18 months, but non-verbal and unable to stand or cruise. He has not had formal neuropsychology evaluation yet due to his frequent hospitalizations”.

line 112: from the photographs seem cutis marmorata without teleangiectasia and in the text you mentioned only cutis mamorata

We have deleted the word “telangiectasia”.

lines141-142: review the number of UCSC genes and OMIM genes. 

Reviewed and added.

line 142: add DECIPHER case and his phenotype. 

There are few cases reported in Decipher. We have mentioned two of them(that overlaps with DNA coordinates in the discussion. We have also included all the Decipher cases as supplementary Table 2 and referred to it in the text.

line 144: actually other Cakut alterations have already been described in the other patients, and these types of issues are heterogeneous even in the same family. 

Corrected and edited.

line148: define FTT

We have previously defined it in the beginning, under section 2. Case report, line 72.

lines 148-149: Additionally, he showed some unique metabolic abnormalities. Define and better explain.

We have provided a better definition of metabolic abnormalities, thank you for your comment.

lines 148-158: CASR is not included in your deletion! 

chr3:121,903,181-122,005,344 

hg 19/GRCh37

this is region of casr and it's outside of your deleted region indicated in the text (123413267_128851482, hg19)

Yes, we have removed this gene and all the sentences about it. Thanks.

lines 159 and 167 : ZNF187, change in ZNF148

We have corrected this.

line163: it's necessary to explain the different types of variants described in Stevens et al. There are also few patients described and so there are limited information. 

This has been addressed as recommended.

line167: explain why znf148 deletion can explain kidney disease. What are the kidney problems described in the patients of stevens et al? how many patients?

This has been added as recommended.

line 176: there are more recent reports like this Homan CC, Venugopal P, Arts P, Shahrin NH, Feurstein S, Rawlings L, Lawrence DM, Andrews J, King-Smith SL, Harvey NL, Brown AL, Scott HS, Hahn CN. GATA2 deficiency syndrome: A decade of discovery. Hum Mutat. 2021 Nov;42(11):1399-1421. doi: 10.1002/humu.24271. Epub 2021 Aug 31. PMID: 34387894; PMCID: PMC9291163. Compare and comments the other gata2 patients in the literature and  and also comment the cancer risk. have you proposed to your patient a cancer surveillance? 

Thank you for alerting us for to this publication. We have added this article to our references and included in our manuscript. Yes, we have discussed the malignancy risk and immunodeficiency with our patient.

lines 179-181: to define a unique features never described before and to hypothesize that they are caused by the deletion, it is necessary to analyze all the clinical differential diagnoses. have you tested Insulinemia? what hepatic echography was? was there familiar hypercholesterolemia? it was caused by food?

We have modified this sentence as follow: “Our subject has additional unreported biochemical features, namely transient hypoglycemia, transaminitis, hypertriglyceridemia and hypercholesterolemia, which could not be fully explained yet based on his clinical history neither the detected chromosomal abnormality”.

line181: add the other OMIM genes in the deleted region and add comments.

We have corrected the gees and added the 6 genes associated with autosomal dominant inheritance.

lines 181: suggest surveillance and exams for future risk. You have not explained what are the clinical problems described in the other patients in the literature and not your patients. 

We have added this as follow:

“Our case continues to have normal complete blood counts (CBC) with differentials at age 27-months and no signs of lymphedema. However, he has recurrent infections, both viral and bacterial with neutropenia but normal immunoglobulins levels. Given the impact of germline GATA2 deletion, surveillance for myeloid malignancy has been initiated in our subject for timely and adequate intervention. We recommend obtaining CBC with differentials every 6-month”. 

line 250: Stevens SJ and not S.SJ. 

let's control the bibliography. 

The citation style used here is Clinical Chemistry style. First name, last name.

Table1:

-you can use abbreviations of years, months and days.

Yes, we did abbreviate them.

- It's not clear who is the last column (patient 2 of Vinh et al)?

Yes. The columns are misaligned. It needs to be adjusted.

-modify the "failure to thrive" part. You have to use or centile or %

Not all papers show the centiles or percents, we can provide for our case

We have indicated that the growth parameters for our patient are all below 1st percentile and we added SD in Table 1 as recommended.

-microcephaly: the patient of Okada doesn't have microcephaly. He presents HC at -0.05 SD. Microcephaly is defined as head circumference ≤-2SD 

We have indicated that the growth parameters for our patient are all below 1st percentile and we added SD in Table 1 as recommended.

Supplementary file: include only the right OMIM genes.

Yes, we corrected these. Thanks.
